# Non-Nutritive Sweetener Intake Is Low in Preschool-Aged Children in the Guelph Family Health Pilot Study

**DOI:** 10.3390/nu14102091

**Published:** 2022-05-17

**Authors:** Anisha Mahajan, Jess Haines, Alex Carriero, Jaimie L. Hogan, Jessica Yu, Andrea C. Buchholz, Alison M. Duncan, Gerarda Darlington, David W. L. Ma

**Affiliations:** 1Department of Human Health and Nutritional Sciences, University of Guelph, Guelph, ON N1G 2W1, Canada; anisha@uoguelph.ca (A.M.); alexjcarriero@gmail.com (A.C.); jaimie.hogan@hotmail.com (J.L.H.); jyu10@uoguelph.ca (J.Y.); amduncan@uoguelph.ca (A.M.D.); 2Department of Family Relations and Applied Nutrition, University of Guelph, Guelph, ON N1G 2W1, Canada; jhaines@uoguelph.ca (J.H.); abuchhol@uoguelph.ca (A.C.B.); 3Department of Mathematics and Statistics, University of Guelph, Guelph, ON N1G 2W1, Canada; gdarling@uoguelph.ca

**Keywords:** preschool, children, non-nutritive sweeteners

## Abstract

There is limited research on the intake of non-nutritive sweeteners (NNS) among preschool-aged children. Canada’s Food Guide suggests limiting intake of NNS for all population groups and Health Canada recommends that young children (<2 years) avoid consuming beverages containing NNS. The aim of this study was to investigate the frequency and type of non-nutritive sweetener (NNS) intake in preschool-aged children participating in the Guelph Family Health Study pilots. Parents (*n* = 78 families) completed 3-day food records (*n* = 112 children; *n* = 55 females, *n* = 57 males; 3.6 years ± 1.3). Nineteen children (17%) reported consumption of foods or beverages containing NNS. Food sources with NNS included: freezies, oral nutritional supplements, flavored water, carbonated drinks, sugar free jam and protein powder. The majority of NNS contained in these foods were identified as stevia leaf extract, acesulfame K, sucralose, monk fruit extract and aspartame. Future research should continue to study NNS intake patterns longitudinally in children and examine the association of NNS intake with diet quality and health outcomes.

## 1. Introduction

Non-nutritive sweeteners (NNS), defined as alternatives to sucrose or high fructose corn syrup, are extracted from natural sources and provide low to no calories [1,2]. These are known by different names such as “low-calorie sweeteners”, “sugar substitutes”, “artificial sweeteners” or “non-nutritive sweeteners” [1,3]. Canada’s Food Guide recommends limiting intake of NNS for all population groups [4]. In addition, Health Canada recommends avoiding the consumption of food or beverages containing NNS by children between birth and 24 months [5]. This is because NNS intake can displace nutrient-dense foods needed for growth and development [3,5]. Currently, NNS intake is being researched in relation to several different health outcomes including gut microbiome, weight management, cardiometabolic health, appetite control, kidney function and dental decay [3]. A systematic review was recently conducted to inform the World Health Organization’s policy and guidelines on NNS use for both adults (≥18 years) and children (<18 years) [6]. This review found that there can be reductions in body weight, BMI Z-score, waist circumference and percent fat mass when NNS were consumed instead of sugar sweetened beverages [6]. This can likely be attributed to a decrease in energy intake with NNS consumption [6]. However, it is important to note that limited research has examined NNS intake in young children (<5 years) [7,8]. A meta-analysis of available randomized controlled trials and cohort studies suggests inconclusive effects of NNS intake in infants and children (<12 years) [9]. 

Understanding intake of NNS in young children is important as there is an increase of NNS in the food supply in North America [10]. This can include foods consumed by children such as candy, yogurts, breakfast cereals, bakery products and jams [4]. In Canada, several NNS are approved for consumption (under acceptable levels) and use by food companies [1,11]. Furthermore, since excessive dietary sugar intake is associated with weight gain, NNS are often promoted as a substitute or “healthier” alternative to sugar intake [11,12,13]. Given that NNS are available in our food supply but are not recommended for young children, it is important to assess and monitor dietary intakes of NNS among young children. Thus, the aim of this study was to examine NNS intake among a sample of children aged 1.5 to 5 years. 

## 2. Methods

### 2.1. Participants

Cross-sectional data from the Guelph Family Health Study (GFHS) were used for these analyses. The GFHS is a family-based health promotion and obesity prevention study that began in 2014 at the University of Guelph, Ontario, Canada [14]. Families with at least one child aged 1.5 to 5 years, who were not planning to move within the next year, were recruited from the Guelph-Wellington area through the local Family Health team, Community Health Centre and Ontario Early Years Centres [14]. This analysis used baseline data collected from pilot families in 2014 and 2015 and was approved by the University of Guelph Research Ethics Board (REB#14AP009). All subjects (parents) gave informed consent to the researchers before participating in the study.

### 2.2. Food Records

Parents completed 3-day paper food records for their children and were given detailed instructions from trained staff to record all food and beverages consumed over 2 weekdays and 1 weekend day, including brand names of products. The paper food records were entered into the ESHA Food Processor Software after which trained staff reviewed every packaged food or beverage to determine if they contained NNS. NNS content and type were determined using product websites and product ingredients lists. Since NNS amounts are not included in the list of ingredients on food product packages in Canada, our study was unable to quantify these amounts. Thus, food records were reviewed for the type of food and beverage items containing NNS, the frequency of intake and type of NNS contained in these products. 

## 3. Results

### 3.1. Demographic Characteristics

The GFHS pilot sample consisted of a total of *n* = 117 preschool-aged children from 83 families. Five children were excluded due to missing or incomplete food records. Among the final analytic sample of *n* = 112 children (*n* = 78 families; *n* = 55 females, *n* = 57 males), the mean age ± SD was 3.6 ± 1.3 years and 82.1 % (*n* = 92) of the participants identified as White. Approximately 47% (*n* = 37) of families reported their annual household income as >$90,000 and 41% (*n* = 32) of families had at least one parent who had completed a university degree (Table 1 and Table 2).

### 3.2. NNS Intake, Frequency and Types

Nineteen (17%) children reported intake of NNS-containing foods or beverages over the 3-day period. Among these children, the intake frequency of foods or beverages containing NNS ranged from 1 to 3 times over the 3-day period. NNS-containing food or beverage items consumed included: freezies, beverages (such as pediatric oral nutritional supplements, flavored water and carbonated drinks), sugar-free jam and protein powder. The main NNS types in these foods were stevia leaf extract, acesulfame K, sucralose, monk fruit extract and aspartame.

## 4. Discussion

Although NNS exposure was limited to a small proportion of children in this study, insights into NNS-containing foods that young children consume were identified. Out of the 112 participants, 19 participants (17%) reported consuming NNS-containing foods. This is higher than NNS intakes from a 2016 Irish study of preschool-aged children (*n* = 500; aged 1 to 4 years), which found that 5% of children consumed NNS [15]. An American study (2009–2012) reported that approximately 25% of children and adolescents (2 to 18 years) were consuming NNS from foods, beverages and packets of NNS [16]. This same study also showed a 200% increase in NNS consumption among children and adolescents from 1999–2000 to 2009–2012 [16]. A 2020 Chilean study found that 65% of children aged 4 to 6 years consumed at least one source of NNS per day [7]. These differing results suggest that the availability of NNS in the food supply could be a key determinant for children’s NNS intake and that NNS intake may be increasing over time. Furthermore, since young children are consuming different kinds of NNS worldwide [2,7,16], further investigation on the impact of NNS intake on diet quality and health outcomes is warranted. 

### Limitations

Although our study addresses an important knowledge gap in current NNS intake patterns among Canadian preschool-aged children, study limitations should be considered when interpreting our results. Specific amounts of NNS in the list of ingredients on food packages were not available, thus, our study was only able to document frequency, and not quantity of NNS intake. Food records are self-reported data and the data for NNS intake, types and frequency were extracted manually by trained staff, which may be subject to error. Dietary intake was assessed using a 3-day food record at a single time point, which may not have sufficiently captured the children’s NNS intake. Assessing dietary intake across more days and at multiple time points could provide a more accurate assessment of NNS intake. Moreover, from the time the data were collected to the time we conducted our final review, certain food product formulations may have changed. Study participant families were primarily White with relatively highly educated parents; thus, our findings may not be generalizable to ethnically diverse or lower socio-economic populations. 

## 5. Conclusions

This is the first study to examine NNS intake in Canadian preschool-aged children. The results indicate that a small percentage of preschool-aged children already consume NNS. Future research is warranted to monitor NNS intake in young children. Furthermore, a major barrier was identified, which is the lack of information about NNS on food labels. Thus, more information on NNS from manufacturers is needed. Given ongoing concerns about NNS intake, additional research should explore the potential effects of NNS intake on diet quality and health outcomes among young children.

## Figures and Tables

**Table 1 nutrients-14-02091-t001:** Characteristics of families (*n* = 78) participating in Guelph Family Health Study pilot studies at baseline.

Characteristic	*N* (%)
Household Income (Canadian) (family data)	
Did not answer—<$39,000	8 (10.2%)
$40,000–79,999	23 (29.4%)
$80,000–89,999	10(13%)
>$90,000	37(47.4%)
Parent Education (family data)	
Postgraduate training or degree	32 (41%)
University or College graduate	38 (48.7%)
Some university, some college or technical school	8 (10.3%)

**Table 2 nutrients-14-02091-t002:** Characteristics of children (*n* = 112) participating in Guelph Family Health Study pilot studies at baseline.

Characteristic	*N* (%)
Child Ethnicity	
White	92 (82.1%)
Other	20 (17.6%)
Child Age in years, Mean ± SD	3.6 ± 1.3 years
Child Sex	
Male	57 (51%)
Female	55 (49.1%)

## Data Availability

The GFHS welcomes external collaborators. Interested investigators can contact GFHS investigators to explore this option, which preserves participant confidentiality and meets the requirements of our Research Ethics Board to protect human subjects. Due to Research Ethics Board restrictions, we do not make participant data publicly available.

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
