# Peer review of "Non-Nutritive Sweetener Intake Is Low in Preschool-Aged Children in the Guelph Family Health Pilot Study"

_nutrients, 2022, doi:10.3390/nu14102091_

Round 1

Reviewer 1 Report

Dear Authors,

The manuscript (nutrients-1679256) submitted for review is interesting as the readers can understand that preschool-aged children also intake consume sweeteners in their diet. I supposed that is impossible. Despite the positive impression that this short communication made on me, I have a few comments.

Authors, Please note the following comments:

Authors should add limitations as a separate section to these results.

A title: I propose to add to the title "Pilot study" because the group of respondents is small, and studies of large populations may show a different result.

Technical notes:

On the first page in the last paragraph, instead of the abbreviation "y", I suggest that the authors should write the full word year.

References

References are not cited according to journal rules. Publications from MDPI provide information on how to properly cite. Authors may also find this information in the authors' guide.

Reviewer

Author Response

Reviewer 1:

Reviewer: Authors should add limitations as a separate section to these results.

Author’s response: Thank you. This has been included in the discussion section as suggested.

Reviewer: title: I propose to add to the title "Pilot study" because the group of respondents is small, and studies of large populations may show a different result.

Author’s response: Thank you. This has been completed.

Reviewer: technical notes: On the first page in the last paragraph, instead of the abbreviation "y", I suggest that the authors should write the full word year.

Author’s response: Thank you. We have changed “y” to “years” throughout the document to keep this consistent.

Reviewer: References are not cited according to journal rules. Publications from MDPI provide information on how to properly cite. Authors may also find this information in the authors' guide.

Author’s response: Thank you. This has been edited as suggested based on this link to align with American Chemical Society: https://mdpi-res.com/data/mdpi_references_guide_v5.pdf

Reviewer 2 Report

Dear Authors,

in my opinion, a 3-day period food records is insufficient for a serious presentation of the NNS intake by preschool-aged children. The time of this examination should be extended  to at least three times on the same group. E.g. three times during three different months, three times during three different weeks. 3 day-long period carried out only once is too little when we want to know the possible NNS intake by the tested group.

Introduction part should be improved with information about the health  risks of NNS consumption in children under the age of 5 years.

Author Response

Reviewer: In my opinion, a 3-day period food records is insufficient for a serious presentation of the NNS intake by preschool-aged children. The time of this examination should be extended to at least three times on the same group. E.g. three times during three different months, three times during three different weeks. 3 day-long period carried out only once is too little when we want to know the possible NNS intake by the tested group.

Author’s response: We thank you for your very insightful comment. At this time, we are unable to change the study design as we have data from one dietary measurement. However, we will keep this in mind for future studies. This limitation has been addressed in the discussion and added as a separate section as suggested by Reviewer 1. However, despite limitations of this study, our pilot work has provided invaluable insight to better understand challenges in the measurement of non-nutritive sweetener (NNS) in Canada and initial estimate of NNS intake in pre-school aged children.  These learnings will help us in future investigations on this topic.

Reviewer: Introduction part should be improved with information about the health risks of NNS consumption in children under the age of 5 years.

Author’s response: As suggested, we have added a few systematic reviews outlining current research on health risks with NNS intake in adults and children. Unfortunately, there are no Canadian studies focused on NNS intake patterns under the age of 5 years and so we have included these into our discussion from an international perspective.